# Impact of *BRCA* Status on Reproductive Decision-Making and Self-Concept: A Mixed-Methods Study Informing the Development of Tailored Interventions

**DOI:** 10.3390/cancers14061494

**Published:** 2022-03-15

**Authors:** Sharlene Hesse-Biber, Memnun Seven, Jing Jiang, Sara Van Schaik, Andrew A. Dwyer

**Affiliations:** 1Department of Sociology, Boston College, Chestnut Hill, MA 02467, USA; 2Elaine Marieb College of Nursing, University of Massachusetts Amherst, Amherst, MA 01003, USA; mseven@umass.edu; 3Department of Measurement, Evaluation, Statistics and Assessment, Boston College, Chestnut Hill, MA 02467, USA; jj910716@gmail.com; 4School of Social Work, Boston College, Chestnut Hill, MA 02467, USA; saravans22@gmail.com; 5William F. Connell School of Nursing, Boston College, Chestnut Hill, MA 02467, USA; andrew.dwyer@bc.edu; 6Harvard Center for Reproductive Medicine, Massachusetts General Hospital, Boston, MA 02114, USA

**Keywords:** *BRCA* mutation, precision health, reproductive decision-making, risk-reducing surgery, self-concept

## Abstract

**Simple Summary:**

Reproductive decision-making is a complex process and is influenced by personal, medical, and sociocultural factors. Relatively little is known about reproductive decision-making in women harboring mutations in the *BRCA1* and/or *BRCA2*—pathogenic variants that confer different cancer risk profiles and underlie hereditary breast and ovarian cancer. A deeper understanding of *BRCA*+ women’s experiences is needed to develop tailored approaches to reproductive decision-making—a central aspect of precision health. This study provides findings to guide tailored interventions to enhance precision health for *BRCA+* women of reproductive age.

**Abstract:**

This mixed-methods study sought to deepen our understanding of self-concept and experiences in balancing cancer risk/reproductive decisions after learning of *BRCA*+ status. First, a quantitative survey of *BRCA*+ women (*n* = 505) examined the childbearing status, risk-reducing surgery, and self-concept. At the time of testing, 307/505 (60.8%) women were of reproductive age (<40 years-old), 340/505 (67.3%) had children, and 317/505 (62.8%) had undergone risk-reducing surgery. A younger age at the time of the testing was significantly associated with the decision to have children after learning *BRCA*+ status or undergoing risk-reducing surgery (*p* < 0.001). Compared to older women, *BRCA*+ women of reproductive age, exhibited a more negative self-concept with significantly higher vulnerability ratings (*p* < 0.01). Women with a cancer diagnosis exhibited a more negative mastery ratings and worse vulnerability ratings (*p* < 0.01) than women without a cancer history. Compared to childless counterparts, significantly higher vulnerability ratings were observed among *BRCA*+ women who had children before learning their *BRCA* status and/or undergoing risk-reducing surgery (*p* < 0.001). Subsequently, a subset of women (*n* = 40) provided in-depth interviews to explore their experiences in decision-making. The interviews provided insights into the effects of *BRCA* status on decisions regarding relationships, childbearing, cancer risk management, and communicating *BRCA* risk to children. Integrating quantitative and qualitative findings identifies targets for tailored interventions to enhance precision health for *BRCA+* women of reproductive age.

## 1. Introduction

Technological advances and growing applications of genomics in clinical practice have increased the availability and utilization of genetic testing for variants in breast cancer 1 and 2 (*BRCA*) genes [1,2]. Genetic testing for *BRCA* genes informs the individual lifetime risk of developing hereditary breast and ovarian cancer (HBOC) and other *BRCA*-related cancers [3,4]. In the general population, the risk of developing breast and ovarian cancer is 12.9% and 1.2%, respectively [5]. Importantly, *BRCA* variants underlie hereditary breast and ovarian cancer (HBOC), and women harboring a pathogenic variant (mutation) in *BRCA* 1 have up to a 72% lifetime cumulative risk of breast cancer and up to 44% risk of ovarian cancer. For women carrying a *BRCA* 2 pathogenic variant, breast and ovarian cancer risks are 69% and 17%, respectively [6,7]. Importantly, pathogenic *BRCA* variants have an autosomal dominant inheritance. Thus, men and women alike have a 50% risk of passing the *BRCA* variant to their offspring [3].

The Centers for Disease Control and Prevention identify HBOC as a ‘Tier 1’ genetic condition meaning that evidence-based guidelines and recommendations support a significant benefit for early detection (i.e., genetic testing) and intervention—with reduced morbidity and mortality. When genetic testing reveals a pathogenic *BRCA* variant, individuals are faced with several complex decisions that are related to their personal cancer risk, how to communicate results to at-risk blood relatives, as well as health decisions (i.e., enhanced surveillance, medical management, and/or risk-reducing surgeries) [8]. Indeed, much of the *BRCA* literature has focused on women’s decision-making regarding genetic testing and risk-reducing surgeries (i.e., mastectomy, salpingo-oophorectomy, hysterectomy). However, learning one’s *BRCA*+ status and the risk of passing the *BRCA* pathogenic variant to offspring may significantly impact reproductive decision-making [1,2,4,8,9].

A 2020 integrative review only identified five articles on reproductive decision-making in *BRCA*+ women [8]. Given the paucity of literature on the subject, study authors highlighted the need for further investigation to better understand how the emotional response to *BRCA* status and women’s values and beliefs influence reproductive decision-making [8]. We, and others, have previously reported a range of emotional reactions and coping responses after the disclosure of a *BRCA*+ status, including confusion, sadness, feeling “blindsided”, uncertainty, anger, denial, and guilt [10,11,12,13]. Prior studies also suggest that women may experience altered self-concept (i.e., an individual’s sense of identity), including feelings of vulnerability, stigma, and decreased self-confidence that impair the health-related quality of life [14]. Such emotional responses and altered self-concept can affect health decisions/behaviors and interfere with planning for the future [15]—including reproductive decisions. Women of reproductive age harboring pathogenic *BRCA* variants have a range of reproductive options, including attempting to conceive naturally (with or without a prenatal genetic diagnosis); utilizing assisted reproductive technologies (ART) with donor sperm/ova, or opting for in vitro fertilization (IVF) with pre-implementation genetic testing/diagnosis; and pursuing adoption, or choosing not to have biological children [8,9]. Importantly, pregnancy is safe and there is no increased risk of adverse maternal/fetal outcomes in women with breast cancer who harbor a pathogenic *BRCA* variant [16]. Further, ART does not appear to increase the risk of cancer recurrence among breast cancer survivors that are harboring pathogenic *BRCA* variants [17]. To date, relatively little is known about *BRCA*+ women’s reproductive decision-making or how counseling and support can be tailored to support high-quality decisions that are informed and aligned with an individual’s values and preferences. Moreover, given the differing cancer risk profiles between *BRCA1* and *BRCA2*, it is unknown whether women’s reproductive decision-making differs between those harboring a *BRCA1* vs. *BRCA2* variant.

Precision health is an emerging approach to individualizing healthcare and improving outcomes for health and wellbeing [18]. Tailored approaches are central to precision health and focus on proactive, personalized solutions to health issues that integrate genetic, behavioral, environmental, and individual lifestyle factors [18,19]. It is widely acknowledged that genetic counseling and reproductive decision-making should be individualized to support informed decisions [8]. Women need to be informed of the potential risks, benefits, and limitations of reproductive options, and comprehensive discussion should elicit a women’s preferences and consider individual goals, needs, value systems, as well as cultural and religious beliefs [8]. As such, precision health can individualize management strategies and approaches to support reproductive decisions for *BRCA*+ women.

We posit that a more comprehensive understanding of *BRCA*+ women’s experiences regarding reproductive decision-making could propel clinical care beyond guideline-based recommendations (designed for populations) towards a more tailored, personalized approach to care that is acceptable and responsive to individual needs [20,21]. Therefore, we aimed to better understand self-concept and reproductive decision-making in *BRCA*+ women to guide the development of tailored approaches to decisional support and promote precision health for women harboring pathogenic *BRCA* variants.

## 2. Materials and Methods

An explanatory, sequential, mixed-methods design was employed because it pro-vides a deeper understanding of how BRCA+ status affects women’s self-concept and reproductive decision-making at different stages in life. Briefly, the quantitative sur-vey examined self-concept and reproductive decisions among BRCA+ women in adulthood. Subsequently, qualitative interviews were conducted with a subset of sur-vey respondents to explore BRCA+ women’s psychosocial experiences and decision-making process in detail.

The study was approved by the Boston College institutional review board, and all the participants provided opt-in electronic informed consent before the initiation of the study procedures. The data are reported according to Strengthening the Reporting of Observational studies in Epidemiology (STROBE) [22].

### 2.1. Participants and Procedures

We used a purposive sample for this mixed-methods study. English-speaking adult women (18+ years-old) harboring a pathogenic *BRCA* variant were included in this study. We purposefully selected women in different reproductive ages and childbearing status. The participants were recruited (January 2015–September 2018) in collaboration with patient support organizations (Facing Our Risk of Cancer [FORCE], National breast Cancer Coalition, Bright Pink) as previously described [11].

First, we used an online quantitative survey to examine *BRCA* self-concept, childbearing status, and risk management strategies (i.e., risk-reducing surgery) after learning of their *BRCA*+ status. Following informed consent, the participants provided socio-demographic data (i.e., age, race, marital status), reproductive information (i.e., childbearing status, number of biologic/adopted children), and details on risk management decisions (i.e., actual or planned risk-reducing surgery). The participants completed the *BRCA* Self-Concept Scale [15]. The 17-item instrument uses a 7-point Likert-type scale to assess self-concept across three subscales: stigma (8 items, α = 0.87), vulnerability (5 items, α = 0.80), and mastery (4 items, α = 0.81). Total and subscale scores are averaged, and higher scores represent a more negative self-concept (α = 0.89). The stigma subscale measures how much women feel stigmatized by their *BRCA* genetic test result and the negative feelings toward themselves (e.g., increasingly secretive, feeling isolated, labeled, burdened, loss of privacy). The vulnerability subscale measures feelings of helplessness concerning their physical condition (e.g., distrust of their body, worries about cancer risk, and passing cancer risk to offspring). The mastery subscale measures self-confidence in dealing with a positive *BRCA* test result and a perceived sense of control of their health [15]. We defined women of ‘reproductive age’ as participants between the ages of 18 and 40. This cut-off was based on the anticipated age-related decline in fertility as well as increased rates of aneuploidy and spontaneous abortion in women after age 40 years [23]. For the quantitative survey, 623 online survey responses were received through patient support organizations. Incomplete surveys and those participants not meeting inclusion criteria were excluded (*n* = 118). In total, 505 participants (81% of respondents) were included in the analyses.

Subsequently, we conducted qualitative interviews to explore women’s ‘lived experiences’ and reproductive decision-making after learning their *BRCA*+ status. A subset of survey participants provided a semi-structured telephone interview that was conducted by a single study investigator (SHB). Although a minimum of 24 interviews is typically needed to reach meaning saturation, we included 40 women to develop a rich understanding of their lived experiences [24]. In-depth telephone interviews (60–120 min) were used to capture the women’s experiences in having a pathogenic *BRCA* variant and their reproductive decisions. The investigator started each discussion by asking the interviewee to “share your *BRCA* story”. Subsequent questions elicited personal medical history—including experiences with genetic testing, disease management (e.g., risk-reducing surgery), and probing reproductive decisions before and/or after learning their *BRCA* status (i.e., satisfaction, distress, regret). The interviews were audio-recorded, transcribed verbatim, and memos were recorded for each interview. Following transcription, the participants were given the option to review their transcribed interview to edit and clarify responses. No participants chose to review their transcripts.

### 2.2. Analysis

Quantitative data were analyzed using the Statistical Package for the Social Sciences (SPSS, V20.0) [25]. The sociodemographic data are reported using descriptive statistics. The normality of the data was assessed using the Kolmogorov–Smirnov test. Bivariate analysis was used to compare the variables between the reproductive age groups (i.e., ≤40 years-old vs. 40+ years-old) by employing the Chi-square test, Mann–Whitney U test, and a Kruskal–Wallis one-way ANOVA with post hoc tests (as appropriate). Pairwise comparisons were created based on age at the time of *BRCA* testing, childbearing status in relation to *BRCA* testing, and childbearing status in relation to risk-reducing surgery. A *p* value < 0.05 was considered statistically significant.

Qualitative interview data were analyzed using template analysis. A relative strength of template analysis is that it follows a structured ‘top-down’ approach using a priori coding templates (termed “themes”) that were identified from the literature or quantitative findings. In the present study, the template themes were drawn from the quantitative survey results of *BRCA*+ women (described above). Template analysis allows additional sub-themes and dimensions to modify and provide depth to the initial template themes (i.e., quantitative findings). The sub-themes and dimensions are identified using a “bottom-up” approach that is based on a line-by-line “grounded reading” of the transcripts [26,27]. Detailed memos for each interview included codes (with corresponding line numbers). Two investigators (SHB, SVS) independently coded interview data and created memos. Emergent analytical themes are identified and evaluated in subsequent interviews. A total of three investigators (SHB, MS, AAD) discussed emergent themes arising from the iterative coding of interview transcripts, and all the investigators agreed to the final coding structure. The iterative coding process enables the identification of emergent sub-codes (representing new analytical ideas) and dimensions (providing additional context and nuance of the sub-codes) to expand on the a priori template themes from the quantitative survey, thereby deepening our understanding of the lived experiences of women harboring pathogenic *BRCA* variants.

## 3. Results

We conducted an explanatory, sequential, mixed-methods study integrating a quantitative survey (*n* = 505) and qualitative interviews (*n* = 40). Women ranged in age from 18 to 70 years-old. In total, 45.1% 228/505 (45.1%) were of reproductive age (i.e., ≤40 years-old). Overall, most women identified as non-Hispanic White, married, college-educated, and employed full-time with an annual household income > USD75,000 (i.e., middle class) (Table 1).

### 3.1. Clinical and Reproductive Characteristics of BRCA+ Women

At the time of *BRCA* testing, 307/505 (60.8%) were of reproductive age, and a third of the women (170/505, 33.7%) had received a cancer diagnosis (i.e., breast or ovarian). The rates of women harboring *BRCA1* and *BRC2* were similar (236/505, 46.7% vs. 259/505, 51.3%, *p* = 0.15). In total, 317/505 (62.8%) women reported having had risk-reducing surgery (i.e., single/double mastectomy, salpingo-oophorectomy). Almost two-thirds of the women (324/505, 64.2%) had biological children (Table 2). Of the women who had risk-reducing surgery, 68.1% (216/317) had biological children. Neither personal history of a cancer diagnosis nor having undergone risk-reducing surgery differed according to the genetic variant (i.e., *BRCA1* vs. *BRCA2*).

### 3.2. Relationship between Timing of BRCA Testing, Childbearing Status, and Reproductive Decisions

Women of reproductive age (18–40 years-old, *n* = 307) at the time of the genetic testing were categorized into one of four groups: (1) the women who had children before *BRCA* testing (*n* = 134, 43.6%); (2) the women who had children after *BRCA* testing (*n* = 18, 5.9%); (3) the women who had children before and after *BRCA* testing (*n* = 30, 9.8%); and (4) the women who had no children (*n* = 116, 37.8%). Significant differences in the reproductive decisions for childbearing were observed across the age groups of women of reproductive age following *BRCA* testing (*p* < 0.001). Notably, few women (16.1%) had children after learning their *BRCA*+ status—regardless of having children before *BRCA* testing. Among younger women (26–30 years-old), most (13/18, 72.2%) had their first child after testing, and 11/30 (36.7%) had another child after learning their *BRCA*+ status. In contrast, few *BRCA*+ women in their late 30 s chose to have children after testing. Younger women were more likely to have another child (or their first child) after testing *BRCA*+ (*p* < 0.001). Moreover, among the women who were childless at the time of *BRCA* testing, only 18/298 (6%) had children after learning their *BRCA* status, and the vast majority of those women (16/18, 88.9%) were under 30 years-old (Figure 1).

### 3.3. Relationship between Risk-Reducing Surgery, Childbearing Status, and Reproductive Decisions

Significant differences in childbearing status were observed across the age groups of women of reproductive age concerning risk-reducing surgery (*p* < 0.001). Of the 236 women of reproductive age who had risk-reducing surgery, 106/236 (44.9%) had children before the risk-reducing surgery, while only 4/236 (1.7%) had children after surgery, and 10/236 (4.2%) had children before and after surgery. A total of 139/236 (58.8%) of women of reproductive age provided information on their specific risk-reducing surgery. Most women of reproductive age who underwent both mastectomy and oophorectomy had children prior to the risk-reducing surgery (39/139, 78%, *p* < 0.05). Only one participant reported having a child before and after oophorectomy. Nearly half of women of reproductive age (116/236, 49.2%) did not have any children. Although significantly fewer women had children after having risk-reducing surgery (106/236, *p* < 0.001), all were younger than 35 years-old.

### 3.4. Self-Concept in BRCA+ Women

Women harboring *BRCA1* (*n* = 236) and *BRCA2* (*n* = 259) were similar in terms of self-concept (total and sub-domains). Scores for ‘stigma’, ‘vulnerability’, and ‘mastery’ subscales did not differ between the women harboring a *BRCA1* vs. *BRCA2* variant. (Table 3). However, age-related differences were noted. The women who tested *BRCA*+ during their reproductive years (<40 years-old) had significantly worse self-concept compared to older women (*p* < 0.05). The scores for ‘stigma’ and ‘mastery’ subscales were similar regardless of age (i.e., ≤40 vs. >40 years-old), childbearing status, or risk-reducing surgery. Women with a personal history of cancer exhibited higher (worse) ‘vulnerability’ subscale scores and lower (better) ‘mastery’ subscale scores compared to women without a cancer diagnosis (Table 3). Moreover, compared to the older *BRCA*+ women, women of reproductive age had significantly higher (*p* < 0.01) ‘vulnerability’ scores (e.g., feelings of distrust in their body, worries about cancer risk, and conferring risk to offspring). Those women of reproductive age who had children prior to *BRCA* testing exhibited significantly higher vulnerability scores than similarly aged, childless *BRCA*+ women (*p* < 0.001). Similarly, there was a trend (*p* = 0.05) towards higher vulnerability scores in women who had children after undergoing risk-reducing surgery compared to their childless counterparts (Table 3).

### 3.5. Qualitative Interviews on Learning BRCA Status and Reproductive Decision-Making

Subsequent qualitative interviews (*n* = 40) were used to enrich, contextualize, and refine the significant findings of the quantitative survey. The representative quotes mapping to the quantitative findings are depicted in Table 4. The quotes depict *BRCA*+ women’s ‘lived experience’ providing context and insight into understanding significant survey findings (i.e., childbearing status at the time of *BRCA* testing, weighing risk-reducing surgery and childbearing, *BRCA*+ status, and self-concept). The women who were single and/or did not have a partner when they learned their *BRCA*+ status described challenges relating to reproductive decisions as well as intimate and/or romantic relationships and marriage. Some women struggled with decisions and felt uncertain about their future prospects for marriage and childbearing (Table 4 Quantitative Survey C, Interviewee #19). Most women who had children before the *BRCA* testing expressed a sense of relief and gratitude that they had reached ‘life milestones’ (i.e., childbearing and breastfeeding) before learning their *BRCA*+ status. However, some women shared their difficulty with reproductive decisions (i.e., to have an additional child) after learning they were *BRCA*+.

One woman considered it selfish even to think about having another child after learning her BRCA+ status: “It was this message that it was very selfish to consider having a third child and then run the risk of getting cancer while I was pregnant or right after the baby was born” (Interviewee #3, 32 yo). Some women were initially traumatized and upset but subsequently found comfort in having already accomplished a ‘life milestone’ (having children) and learning their *BRCA* status did not alter their reproductive plans (Table 4 Quantitative Survey A, Group 1 quote). Women who had children prior to *BRCA* testing conveyed confidence in managing their cancer risk (i.e., medical/surgical interventions) with little concern for how treatment decisions might affect childbearing plans. However, the women who had children before testing described feelings of ‘parental guilt’ related to having possibly transmitting their pathogenic *BRCA* variant to their child(ren). Most women were preoccupied with the implications of *BRCA* on their children—specifically their daughters. Women were very distressed and felt unsure how to communicate their *BRCA* status to their children and explain how *BRCA* might affect different aspects of their child(’s) life (e.g., marriage and childbearing) (Table 4 Quantitative Survey C, Interviewee #3).

Upon learning their *BRCA*+ status, women had to weigh cancer risk management decisions (i.e., risk-reducing surgery)—interventions that could compromise fertility and/or significantly affect future childbearing plans. Several younger women felt ‘jarred’ and ‘shocked’ after learning their *BRCA* status as they had not yet seriously considered family planning or reproductive decisions and were now forced to do so. A 25-year-old woman stated: “*I kept crying because all summer… I was like, ‘My life is over*’” (Interviewee #10).

Reproductive decision-making presented different challenges and considerations for women who had children before testing and had another child(ren) after learning their *BRCA*+ status. Women were generally grateful for having at least one child, and some hoped to “complete” their family before pursuing risk-reducing surgery. One 34-year-old woman found herself weighing both risk management decisions and reproductive options in the wake of learning her *BRCA*+ status (Table 4 Quantitative Survey B, Group 1 quote). Childless women (no children before or after *BRCA* testing) shared different ‘lived experiences’ regarding their reproductive decision-making. Although some women stated a *BRCA*+ test result confirmed/solidified their desire not to have children, others felt compelled to not pursue conception for fear of passing the pathogenic variant to their child(ren) (Table 4 Quantitative Survey A, Group 2 and 4 quotes). One 25-year-old interviewee stated: “*No, absolutely not, I don’t [want children]. I do not want my child to have to deal with the things that I’m dealing with. I don’t want there to be any possibility that I pass this [BRCA] on to them. One, because I’d feel like sh*t; two, because then they would have to deal with all of this; and three, because then there would be a chance that one day, they would pass it [BRCA] on*” (Interviewee #10).

### 3.6. Qualitative Interview Results on Risk-Reducing Surgery and Reproductive Decision-Making

Among the women who underwent oophorectomy, 17/35, (48.6%) had a child(ren) before surgery or did not have any children (17/35, 48.6%). Only one woman reported having a child before and after oophorectomy. Although many women appreciated the option to mitigate the cancer risk, the decision to undergo risk-reducing surgery included many considerations, negotiations, and conflicting thoughts concerning reproductive decisions. The interviews revealed that the women who had not ‘completed’ their childbearing and who planned/desired to experience breastfeeding were particularly distressed (Table 4 Quantitative Survey B, Group 3 quote). Among the older *BRCA+* women of reproductive age, most felt conflicted about their cancer risk (i.e., delaying risk-reducing surgery and choosing to conceive again). A 35 year-old woman stated: “*Between [my and my husband’s BRCA+ status] and my age, there are also some implications there for whether we have a second child… So it’s a little aggravating to [go through] the fertility tests… I don’t know if that’s an option we want to pursue. Cause I honestly don’t know if it’s wise for someone who’s at risk for ovarian cancer to hyper-stimulate their ovaries… I’m not against having another child per se, but the procedures to do that [are] varying*.” (Interviewee #14).

Several women felt grateful for having children after risk-reducing surgery—yet expressed sadness and regret about not being able to experience ‘life milestones’ such as breastfeeding: “*Every now and then I am reminded like… oh, that they [breast implants] are still not like a real part of my body. That was something else that I had to like mourn and come to terms with, you know, having my daughter after all of this and not having that option to breastfeed or, you know, anything like that.”* (Interviewee #8). The tension between concerns for one’s health (i.e., mitigating cancer risk) and envisioned goals (i.e., ‘life milestones’ of childbearing/breastfeeding) created a sense of a ‘compressed timeline’. Such pressure in balancing cancer risk and reproductive planning was reflected by Interviewee #10: “*I feel like, you know, rushing into a relationship and a marriage and having kids in the next three years with someone I’m not sure about.... it would be worse than getting breast cancer right now… because you do that and you survive, but then you’re living this life that’s not the one that you would’ve lived… and, I don’t know that’s better… like, if I.., you know, the other outcome is that I wait too long and I… and I have to get surgery without having kids and then…, you know, and then what*?”.

### 3.7. Self-Concept Related to BRCA Status, Childbearing, and Risk-Reducing Surgery Decisions

Expanding on the quantitative survey findings of heightened feelings of vulnerability, the interviews revealed intimate descriptions of vulnerability in relationships, making reproductive decisions, and health risks (Table 4 Quantitative Survey C, Interviewee #19). A woman who underwent a mastectomy before attempting to get pregnant (with difficulty) described her experience as “*just a kick in the teeth after going through everything*” (Interviewee #8). The interviews point to the stigma that is associated with learning *BRCA*+ status, including a sense of being burdened with knowing their *BRCA* status and feeling that being *BRCA*+ got in the way of being “who I really am”. The interviews revealed that many women considered motherhood (i.e., having children and getting to breastfeed them) as an expectation and ‘life milestone’. When *BRCA* status compromised fertility/breastfeeding plans, women perceived it as a significant distressing and burdensome life event.

### 3.8. Explanatory Findings Integrating Quantitative and Qualitative Results

As depicted in Table 4, the qualitative interviews provide insight and context for understanding the significant quantitative (survey) results. A significant survey finding was the increased ‘vulnerability’ (particularly in women of younger reproductive age) that served as the template theme for the qualitative analysis of interview data. Figure 2 depicts a schematic that visually presents the explanatory findings of this mixed-methods study and synthesizes the data that are presented in Table 4. After learning their *BRCA*+ status, women of reproductive age faced a matrix of factors affecting their reproductive decision-making that helped elucidate significant survey results. Both younger and older women shared experiences of bargaining or negotiation that played out over a developmental timeline. Notably, *BRCA*+ women were challenged by conflicting concerns for self and others (i.e., family, children).

## 4. Discussion

Herein, we report findings of a mixed-methods study that sought to deepen our understanding of *BRCA*+ women’s self-concept and reproductive decision-making. In our sample, most women (61%) were of reproductive age (18–40 years-old) when they underwent *BRCA* testing and two-thirds (68%) had a child at the time of testing. Our observations are similar to a recent study reporting 54/139 (39%) *BRCA*+ women were younger than 35 years-old at the time of testing and 99/139 (71%) had a child at the time of testing [2]. As *BRCA* testing expands, there are growing numbers of women of reproductive age who harbor pathogenic *BRCA* variants [28]. Further, HBOC is a Tier 1 condition and genetic testing for *BRCA* variants provides significant benefits for mitigating cancer risk through risk-reducing surgeries and/or increased surveillance. Discovering a pathogenic *BRCA* variant is generally considered a watershed, life-changing experience. We observed an altered self-concept in *BRCA*+ women and the impact of *BRCA*+ status did not differ between women carrying a *BRCA1* mutation and those harboring a pathogenic *BRCA2* mutation. For women of reproductive age, a positive test result initiates a cascade of decisions regarding relationships, reproduction, and childbearing [9,28,29]. Yet to date, there is little literature on reproductive decision-making to tailor approaches to needs and preferences of *BRCA*+ women of reproductive age.

In the present study, only a small proportion (6%) of the women of reproductive age had children after learning their *BRCA* status. Moreover, our quantitative findings revealed an apparent, significant effect of age on reproductive decision-making. Younger women were more likely to have a child after testing. Further, younger women had significantly higher ratings of ‘vulnerability’, suggesting a strong undercurrent of uncertainty after discovering that they carry a pathogenic *BRCA* variant. Subsequent qualitative interviews provided context for the quantitative findings (Figure 2) and identified *BRCA*+ women had expectations of reaching particular ‘life milestones’ (i.e., childbearing/breastfeeding). Such expectations were important drivers of reproductive decision-making in *BRCA*+ women—yet women often struggled with conflicting feelings in balancing cancer risk management and reproductive goals as well as feelings of ‘vulnerability’ in relationships. In one of the few studies on this subject, Haddad et al. included *BRCA*+ women that were younger than 35 years-old at the time of testing who did not have a personal history of breast or ovarian cancer. The investigators found that women were more likely to report feelings of urgency to have a family after learning their *BRCA* status [2]. Reproductive decisions are complex and are affected by various physical, psychological, social, cultural, moral/ethical motives, and considerations [30]. In the present study, women conveyed several dilemmas, including personal reproductive concerns (i.e., assisted reproductive technology—in vitro fertilization with pre-implementation genetic testing) and concerns for offspring (i.e., not wanting to confer *BRCA* risk to children). Although, ART does not appear to increase risk of cancer recurrence [16,17], some women shared fears of increased cancer risk with IVF during interviews. In addition to patient knowledge gaps, physicians that are involved in cancer care also have misconceptions regarding fertility preservation and pregnancy-related issues in women with breast cancer [31]. Thus, tailoring counselling to address such misconceptions is critical for those women who are considering pregnancy after genetic testing. The fear of passing cancer risk on to children was similarly highlighted in our interviews with older *BRCA*+ women. Thus, traditional genetic counseling and psychosocial support relating to at-risk blood relatives are critical for all women regardless of age [11,12]. Indeed, a recent systematic review and meta-analysis revealed that family-based interventions are needed to support intra-familial communication of cancer risk [32].

Concerns that were related to hereditary cancer risk also appeared to affect women who were childless at the time of genetic testing. Many childless *BRCA*+ women opted not to conceive after learning their *BCRA* status. For some, the test result confirmed their desire not to have a child, but for many, it was a ‘forced’ decision due to concerns of passing the pathogenic variant on to offspring. The fear of transmission was particularly salient for the women of younger reproductive age. Indeed, women with children were primarily focused on the uncertainty about having passed the variant to their child(ren) and the unknown impact it may have on their child’s life if they were found to have inherited the pathogenic *BRCA* variant.

Most women (62%) in our sample underwent risk-reducing surgery (i.e., mastectomy, oophorectomy), and many did so in their 30 s and 40 s. Women in earlier years of reproductive age were more likely to have children after risk-reducing surgery regardless of having children before learning their *BRCA*+ status. The loss of ‘life milestones’ was reflected in qualitative interviews findings that underscored the ‘finality’ of childbearing/breastfeeding following risk-reducing surgery. Women who were later in their childbearing years shared a sense of racing against a ‘cancer clock’ that was accompanied by pressure and urgency to have children as soon as possible to ‘complete’ their family prior to risk-reducing surgery. Our findings echo a 2013 qualitative study of 25 women (with/or without a personal cancer history) who similarly reported a sense of urgency to have children prior to risk-reducing surgery—particularly among childless *BRCA*+ women [4].

Prior studies suggest that uncertainty following a positive *BRCA* test result can have detrimental effects on a woman’s sense of identity (i.e., self-concept, self-esteem, ego integrity). Specifically, uncertainty about personal cancer risk/consequences and familial concerns (i.e., worry about passing/having passed the variant to children) may affect self-identity and self-esteem [15,29,33]. In a study of 237 *BRCA*+ women, Vodermaier et al. examined how self-esteem, mastery, and perceived stigma affect long-term adjustment (up to eight years) following genetic testing. Regardless of the time since the testing, a younger age was associated with greater perceived stigma among *BRCA*+ women (with/without a personal cancer history) [29]. Notably, scores for ‘stigma’, ‘vulnerability’, and ‘mastery’ subscales did not differ in the present study between women harboring a pathogenic BRCA1 vs. BRCA2 variant. This observation suggests that while BRCA1 and BRCA2 pathogenic variants confer different cancer risks, the impact on women’s self-concept appears to be similar. However, BRCA+ women of reproductive age, had significantly worse self-concept scores than older *BRCA*+ women—the differences that were driven by ‘vulnerability’ subscale scores (e.g., feelings of distrust in their body, worries about cancer risk, and conferring risk to offspring). Further, ‘mastery’ and ‘vulnerability’ scores differed according to personal history of cancer–perhaps reflecting a certainty vs. not knowing if cancer was looming. Prior work has posited that a cancer diagnosis may cause women to perceive their body as a threat (i.e., cancer-related danger), requiring women to reframe their self-identify as a patient [34]. Discovering a pathogenic *BRCA* variant after a cancer diagnosis may not worsen women’s self-confidence or perceived control over their health. However, it may increase a sense of ‘distrust’ in one’s body as well as concern about passing cancer risk to offspring. Notably, a sense of mastery is associated with lower general distress among women at risk of HBOC [33]. Thus, it appears critical to tailor resilience-building interventions for women to bolster their perceived mastery and increase self-confidence in managing their health.

Our qualitative interviews revealed ‘vulnerability’ stemmed from feeling overwhelmed by the multiple competing decisions relating to relationships/marriage, managing cancer risk, and reproductive decisions—creating a sense of urgency and a ‘race against the clock’. Similarly, Haddad et al. reported that a positive *BRCA* test result significantly impacts the romantic relationships of younger women [2]. Moreover, in our study, the women of reproductive age who had a child(ren) before *BRCA* testing exhibited higher vulnerability compared to similarly aged, childless *BRCA*+ women. Similarly, *BRCA+* women who had children after risk-reducing surgery had increased ‘vulnerability’ compared to their childless counterparts. Thus, heightened vulnerability may reflect concerns regarding passing a mutated gene to children but also an altered body image and loss of ‘life milestones’ (i.e., not being able to breastfeed). The vulnerability that is experienced by women with children most often centered on concern for their child(ren), i.e., when and how to tell them, consequences on their future relationships, and childbearing plans/decisions. Concerns that were expressed in interviews were particularly focused on daughters. This observation points to gaps in genetic literacy—as sons and daughters are equally likely to inherit a *BRCA* variant from their parents, and pathogenic variants confer increased cancer risk in both males and females. Such misconceptions may contribute to “parent of origin” differences in *BRCA* outcomes [11,12].

Of note, a recent study from the Netherlands examined the use of an online patient decision-aid to support 131 *BRCA*+ women in reproductive decision-making [30]. After three months following the patient decision-aid intervention, the investigators found significant positive effects such as increased *BRCA* knowledge, realistic expectations, lower deliberation, and decreased decisional conflict—and nearly 60% had made an ‘informed decision’. The investigators concluded that the online decision-aid was an appropriate complement to standard reproductive counseling. Future work may involve the assessment of ‘patient centeredness” of patient decision-aids, i.e., that decisions are not only well-informed but also aligned with the individual’s values and preferences. A patient-centered tool—that responds to the themes emerging from the qualitative interviews that are reported here—would represent a significant advance for precision healthcare for women harboring pathogenic *BRCA* variants.

Based on our survey and qualitative findings, we propose that decisional support should include three key components. First, women need clear and understandable information on cancer risk and reproductive options. We envision that a psycho-educational intervention could be a key aspect of supporting informed decisions for *BRCA*+ women. Second, findings suggest a lifespan perspective can be useful to tailor person-centered ‘precision’ counseling and support (Figure 2). We envision that approaches to women of reproductive age focus on personal risk (i.e., rebuild trust in their body) and emotional support for navigating competing cancer and reproductive decisions on a compressed timeline (i.e., ART, sperm/ova donation, IVF with pre-implantation genetic testing). For older *BRCA*+ women, emphasis should focus on supporting concerns for family/children, alleviating parental guilt, and provide skill-building exercises to increase confidence in intrafamilial communication of risk. Women with children may also benefit from understanding actions that can be taken to address potential risk for children who may have inherited the pathogenic *BRCA* variant. Third, discussion should elicit needs, values, cultural/religious beliefs, and preferences for cancer treatments (i.e., risk-reducing interventions) and reproductive goals followed by an opportunity for reflection to support high-quality decisions that are informed and aligned with the individual’s values and preferences.

### Study Limitations and Future Directions

The relative strengths of this mixed-methods study include the sizeable sample (qualitative: *n* = 505, quantitative: *n* = 40) and rigorous template analysis (iterative transcript review, independent coding, detailed memos). The study limitations include a risk of ascertainment bias as women were recruited in collaboration with *BRCA* patient support organizations. The sample was relatively homogeneous (i.e., mostly well-educated, middle-class, non-Hispanic White women), limiting our findings’ generalizability to more diverse BRCA+ populations. In addition, we focused exclusively on women with BRCA variants-yet men can also carry BRCA variants. Future work should include more diverse, representative samples in regards to race and ethnicity as well as sex, gender identity and sexual orientation (i.e., not just cis-gendered heterosexual women). 

There are several future directions that could deepen our understanding of re-productive decision-making related to BRCA carrier status. In the present study we de-fined women of reproductive age as women from 18-40 years of age based on the re-productive endocrine literature. However, it is worthwhile to note that age 40 year is not a definitive cut-off for reproduction given advances in ART. Due to the cross-sectional study design, data collected at a single timepoint that may not reflect the dynamic nature of self-concept. Self-concept is shaped based on one’s life experi-ences and may change over time. Accordingly, longitudinal studies are needed to ex-amine how self-concept may evolve over time following the revelation of BRCA+ sta-tus. Attitudes of partners may influence reproductive decision-making and future studies could examine dyadic processes as well as patient centeredness of decisional support for BRCA+ women. In addition, ART is becoming increasingly widespread and we did not specifically assess BRCA+ women’s attitudes regarding ART or preimplan-tation genetic testing. Additional work is needed to understand these aspects of re-productive decision-making for BRCA+ women.

## 5. Conclusions

*BRCA*+ women of reproductive age exhibit altered self-concept and high levels of perceived vulnerability. Women with a personal cancer history exhibit greater perceived mastery, yet high levels of vulnerability. Compared to older *BRCA*+ women, women of younger reproductive age were more likely to have children after *BRCA* testing. Our quantitative findings suggest a strong role of developmental life stage in the reproductive decision-making with childbearing/breastfeeding as key ‘life milestones’ that provide meaning for women and help shape self-concept. Qualitative interviews provided depth and context, revealing significant distress and a matrix of life stage factors that affect reproductive decision-making, including childbearing status, uncertainty (i.e., relationship, cancer, fertility), and hereditary concerns of passing the *BRCA* variant to offspring. *BRCA*+ women shared intimate lived experiences that can guide the development of more tailored psychosocial support (e.g., patient decision aids) to support active coping response, promote self-efficacy, and build resilience. Integrating quantitative and qualitative findings highlights the need for more tailored, theory-informed interventions to support reproductive decision-making of *BRCA+* women of reproductive age—who appear to be the most vulnerable and experience the greatest distress concerning reproductive decisions. Specifically, tailored, person-centered approaches are needed to support younger women in navigating decisions on dating/marriage, weighing decisions regarding timing of risk-reducing surgeries reproductive options, and promoting active coping in response to the potential loss of achieving life milestones.

All *BRCA*+ women—regardless of age or childbearing status—merit support from genetic counselors and other healthcare providers to enable effective intra-familial risk communication to mitigate cancer risk and improve health outcomes.

## Figures and Tables

**Figure 1 cancers-14-01494-f001:**
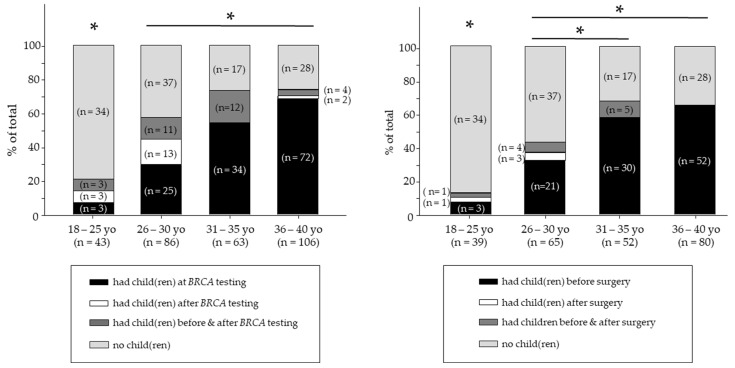
Childbearing status in *BRCA*+ women at the time of testing (left panel, *n* = 298) and risk-reducing surgery (right panel, *n* = 236). The childbearing status among 18–25 year-old women differed significantly from other age groups. Bars depict comparisons, * denotes *p* < 0.001.

**Figure 2 cancers-14-01494-f002:**
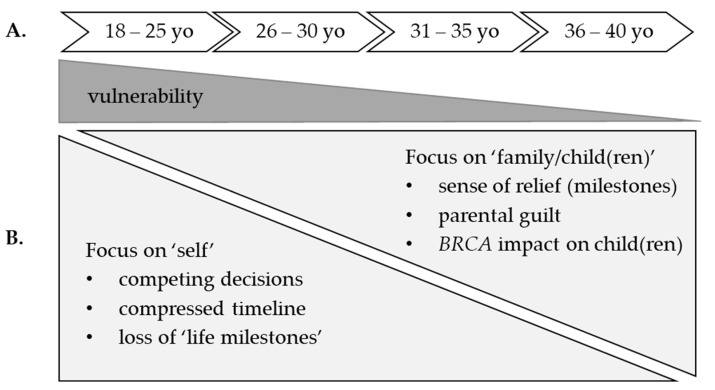
Schematic depicting the template theme, sub-themes, and dimensions in women of reproductive age (≤40 year-old [yo]). (**A**) The template theme from the quantitative findings related to increased ‘vulnerability’ in women of reproductive age. Vulnerability (dark tapered triangle) was the highest in younger women but still present in older reproductive-aged women. (**B**) The sub-themes (triangles) represented a negotiation or bargaining between ‘self’ and ‘others’. Younger *BRCA*+ women tended to focus on the ‘self’, including dimensions (bullets) relating to: (i) balancing cancer risk and fertility goals, (ii) feeling a sense of urgency to make major life decisions (racing against a “cancer clock”) and (iii) concerns about possibly not being able to have children/breastfeed. Older *BRCA*+ women tended to focus on ‘family/child(ren)’ including dimensions relating to: (i) feeling relieved at having met ‘life milestones’, (ii) guilt for possibly passing *BRCA* to offspring and (iii) concern about the impact *BRCA* might have on their child(ren)’s life.

**Table 1 cancers-14-01494-t001:** Sociodemographic of the study participants.

	Quantitative Survey (N = 505)	Qualitative Interview (N = 40)
	*n* (%)	*n* (%)
Age at the time of study		
18–25 yrs.	19 (3.8%)	8 (20%)
26–30 yrs.	42 (8.3%)	5 (12.5%)
31–35 yrs.	71 (14.1%)	8 (20%)
36–40 yrs.	96 (19%)	7 (17.5%)
41–50 yrs.	156 (30.9%)	7 (17.5%)
51–60 yrs.	86 (17%)	3 (7.5%)
61–70 yrs.	35 (6.9%)	2 (5%)
Race and ethnicity		
White/Caucasian	474 (93.9%)	35 (87.5%)
Hispanic and/or Latino	6 (1.2%)	1 (2.5%)
Black/African-American	4 (0.8%)	-
Asian/Asian-American	5 (1.0%)	-
American Indian/Alaska Native	2 (0.4%)	-
Mixed/Other	12 (2.4%)	4 (10%)
Marital status		
married	334 (66.1%)	26 (65%)
single	170 (33.7%)	14 (35%)
not reported	1 (<1%)	-
Education		
high school	47 (9.3%)	-
some college	292 (57.8%)	26 (65%)
college/advanced degree	157 (31.1%)	14 (35%)
not reported	9 (1.8%)	-
Employment		
full time	298 (59%)	30 (75%)
part-time	85 (16.8%)	-
unemployed	73 (14.5%)	2 (5%)
student	17 (3.4%)	6 (15%)
retired	31 (6.1%)	2 (5%)
other	1 (<1%)	-
Household income (annual)		
>$126,000	127 (25.1%)	6 (15%)
$76,000–125,000	140 (27.7%)	18 (45%)
<$75,000	208 (41.1%)	4 (10%)
not reported	30 (5.9%)	12 (30%)

**Table 2 cancers-14-01494-t002:** Clinical and reproductive characteristics of *BRCA+* women (*n* = 513).

Characteristics	*n* (%)
Personal history of cancer	
yes	170 (33.7%)
no	290 (57.4%)
not reported	45 (8.9%)
Age at the time of the genetic testing	
Of reproductive age (≤40 yrs.)	307 (60.7%)
18–25 yrs.	43/307 (14%)
26–30 yrs.	87/307 (28%)
31–35 yrs.	69/307 (22%)
36–40 yrs.	108/307 (36%)
Non-reproductive age (40+ yrs.)	198 (39.2%)
41–50 yrs.	133/198 (67.1%)
51–60 yrs.	50/198 (25.2%)
61–70 yrs.	15/198 (7.5%)
Pathogenic *BRCA* variant	
* BRCA*1	236 (46.7%)
* BRCA*2	259 (51.3%)
* BRCA*1 & *BRCA*2	10 (2%)
History of risk-reducing surgery *	
yes	317 (62.8%)
no	97 (19.2%)
other surgery (not risk-reducing)	39 (7.7%)
not reported	52 (10.3%)
Children	
biological child(ren)	324 (64.2%)
adopted child(ren)	16 (3.2%)
no children	165 (32.7%)

* single/double mastectomy, oophorectomy, hysterectomy.

**Table 3 cancers-14-01494-t003:** *BRCA* self-concept scale scores in *BRCA+* women (*n* = 505).

	Stigma (Mean ± SD)	Vulnerability (Mean ± SD)	Mastery (Mean ± SD)	Total (Mean ± SD)
Of reproductive age at the time of *BRCA* testing				
≤40 years-old (*n* = 312)	3.20 ± 1.2	4.58 ± 1.4	5.36 ± 1.1	4.11 ± 0.8
40+ years-old (*n* = 201)	3.06 ± 1.4	4.22 ± 1.4	5.36 ± 1.1	3.95 ± 0.8
Z (*p* value)	−1.41 (0.15)	−2.75 **(0.006)**	−0.01 (0.99)	−2.23 **(0.02)**
Personal history of cancer				
yes	3.11 ± 1.3	4.62 ± 1.4	5.20 ± 1.1	4.05 ± 0.8
no	3.17 ± 1.3	4.32 ± 1.4	5.46 ± 1.1	4.05 ± 0.8
Z (*p* value)	−0.50 (0.61)	−2.28 **(0.02)**	−2.84 **(0.005)**	−0.18 (0.85)
Pathogenic *BRCA* variant				
* BRCA1*	3.14 ± 1.3	4.38 ± 1.5	5.40 ± 1.2	4.03 ± 0.8
* BRCA2*	3.19 ± 1.3	4.50 ± 1.3	5.33 ± 1.1	4.08 ± 0.8
* BRCA1* & *BRCA2*	2.80 ± 1.2	4.33 ± 1.3	5.25 ± 0.9	3.83 ± 0.7
H (*p* value)	0.71 (0.70)	0.36 (0.83)	1.52 (0.46)	0.99 (0.60)
Childbearing status and *BRCA* testing ^†^				
child(ren) before testing	3.12 ± 1.3	4.61 ± 1.4	5.32 ± 1.1	4.08 ± 0.8
child(ren) after testing	3.49 ± 1.3	4.97 ± 1.3	5.41 ± 1.2	4.37 ± 0.7
child(ren) before & after testing	2.95 ± 1.3	4.56 ± 1.4	5.34 ± 1.1	3.97 ± 0.8
no children	3.23 ± 1.3	4.14 ± 1.4	5.39 ± 1.2	4.00 ± 0.8
H (*p* value)	2.22 (0.52)	11.58 **(0.009)** ^a^	0.63 (0.88)	3.47 (0.32)
Childbearing status and risk-reducing surgery ^††^				
child(ren) before surgery	3.00 ± 1.3	4.59 ± 1.4	5.33 ± 1.1	4.02 ± 0.8
child(ren) after surgery	3.08 ± 1.8	4.73 ± 1.2	6.00 ± 0.6	4.25 ± 0.9
child(ren) before & after surgery	3.23 ± 0.9	4.54 ± 1.0	5.37 ± 1.1	4.12 ± 0.5
no children	3.23 ± 1.3	4.14 ± 1.4	5.39 ± 1.2	4.00 ± 0.8
H (*p-*value)	3.13 (0.37)	7.81 **(0.05)**	1.07 (0.78)	0.61 (0.89)

Significant results are noted in bold text. ^†^ Total *n* = 480; children before testing *n* = 267, after testing *n* = 18; before & after *n* = 30; no children *n* = 165. ^††^ Total *n* = 376; children before surgery *n* = 197, after surgery *n* = 4; before & after *n* = 10; no children *n* = 165. ^a^ Significant difference only between ‘children before testing’ and ‘no children’.

**Table 4 cancers-14-01494-t004:** Integration of quantitative and qualitative findings.

Quantitative Survey Significant Findings	Qualitative Interviews Representative Quotes Depicting Women’s ‘Lived Experience’
**A. Childbearing status at the time of *BRCA* testing**	
Women of reproductive age comprised four groups: (1) had children before testing (2) had children after testing (3) had children before & after testing (4) had no children Significantly more women were of reproductive age when they underwent *BRCA* testing. Most *BRCA*+ women had children before testing. Most women of reproductive age with children opted not to have children after learning their *BRCA* status. Women earlier in their reproductive years were more likely to have children after learning their *BRCA* status.	Group 1: *“I was pretty upset for a few months, but I was also like ‘I’m done having my kids, I’ve done that. I’m done, I’ve nursed my kids… my babies’. You know, they’re growing up… I’m done with that part of my life.”* (Interviewee #5) *“I like always hated kids, but now... How do I know five years from now? I’m not gonna want a kid or have kids… Like a lot of my friends are older than me, and they are getting engaged, getting married, or they’re having kids. And I’m like crap, like, maybe I do want to do that... I feel like I would be thinking about that stuff a lot later if I didn’t know about BRCA.”* (Interviewee #1) Group 2: *“I’ve had friends that have said, ‘Well, I can’t believe that you’d want to have kids with that [BRCA].’ And I’m like…it kind of shocks me sometimes because I’m like, ‘Really’? You think this is the worst thing I could have?’ I kind of think that you know…I look at some of the other things you could have, and I’m like, ‘I’ll take my BRCA, and I’ll be happy about it.”* (Interviewee #15) Group 3: *“I didn’t know if I would have had a third child or not. I think going through this process… it was ultimately my choice to have another one, and I knew when I was turning thirty-five, I wasn’t gonna have that choice anymore. So, we… we ended up um… having another child eighteen months later…I always wanted a bunch of kids.”* (Interviewee #6) Group 4: *“My husband and I have been together for about 13 years now, and neither of us had ever had the inkling to want children in our entire life. We already knew, even before we found out about [BRCA], that we weren’t going to have kids. It kind of seals the deal for us, knowing that I could pass this on to them, and I definitely wouldn’t want to do that. So, it kind of helped solidify the decision not to have children for us…. I want to do a hysterectomy.”* (Interviewee #20)
**B. Weighing risk-reducing surgery and childbearing**	
Women of reproductive age who underwent surgery comprised four groups: (1) had children before surgery (2) had children after surgery (3) had children before & after surgery (4) did not have children Women who were earlier in their reproductive years were significantly more likely to have children after risk-reducing surgery (or before and after risk-reducing surgery).	Group 1:*“For me, I always knew that I would probably test [BRCA], but I had… it had always been in the back of my mind, and I wanted to finish my family first. Because I didn’t know necessarily what steps I would want to take once I found out. But at the same time, it was like… I have two kids that I have to worry about now that I need to figure out … My husband and I had a million and one conversations, and we decided to try for another child... If we didn’t get pregnant … [then] I was going to figure out my surgical option. So, we [had] another baby … I had my salpingectomy at six weeks post-partum. At thirty-four, I’m not ready to go through menopause, and so I’m going to breastfeed this baby until she’s six months old … and I’m hoping to do my mastectomy in December.”* (Interviewee #12) Group 2: *“My boyfriend and I... have a ring. We’re getting engaged in the next few weeks. So, I want to get married, and I know I want to have one kid. I’ve always felt that way. I just want to have one kid. So when I get that… in like, the next three to four years… not out of the way, but for lack of a better term. I then think that I will go ahead and have the surgery. So, sometime in my thirties, my early thirties… I want to have the surgery.”* (Interviewee #13) Group 3: *“That [not being able to breastfeed] was something else that I had to… like, mourn and come to terms with. You know? Having her [another child] after all of this and not having that option to breastfeed.”* (Interviewee #8) Group 4: *“And then, eventually I realized… you know what I mean? If I’m not going to have a baby… why? And with my horrible history… why not do it [surgery] you know?”* (Interviewee #7)
**C. *BRCA+* status and self-concept**	
Women of reproductive age had altered self-concept with significantly higher ratings of ‘vulnerability’. Women of reproductive age with children at the time of *BRCA* testing exhibited significantly higher ratings of ‘vulnerability’ compared to similarly aged, childless *BRCA+* women. Women with a personal cancer history exhibited significantly better ratings of ‘mastery’ and worse ratings of ‘vulnerability’ than women without a personal cancer history.	*“I feel like there’s multiple ticking time bombs. I feel like there’s a ticking time bomb related to my fertility and when I need to… need to meet someone, get married, have children.” (*Interviewee #19*)* *“I have no idea how to tell them [daughters]. What I do know is, I don’t want them [doctors] testing them [daughters] until they’re in their thirties, because I don’t want them [daughters] picking these guys that are the wrong guys and having children early because they’re worried about it [BRCA]” (*Interviewee #3*)* *“Thinking about, um, when to tell her [daughter]. When… when to push her to get tested? Things like that. It’s really about… like, when do you tell them? It’s almost like… you know, adopting a child. When do you tell them something like this?” (*Interviewee #2*)*

## Data Availability

De-identified data will be made readily available upon request for research purposes to qualified individuals within the scientific community.

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
