# Peer review of "Impact of BRCA Status on Reproductive Decision-Making and Self-Concept: A Mixed-Methods Study Informing the Development of Tailored Interventions"

_cancers, 2022, doi:10.3390/cancers14061494_

Round 1
Reviewer 1 Report
The paper by Hesse-Biber and colleagues explores the complex and underreported challenges women with BRCA mutations face when considering reproduction. The sample is sizeable, and the mixed method of analysis ensures both quantitative and qualitative aspects to be valued.
A few minor considerations/explanations could in my opinion improve the message delivered.
A few additional papers clarifying pregnancy outcomes in BRCA mutated patients could help setting the scene and facilitating the disruption of one of the several taboos in this population of patients (e.g., Pregnancy After Breast Cancer in Patients With Germline BRCA Mutations. Lambertini M et al. Clin Oncol. 2020 Sep 10;38(26):3012-3023; Knowledge, attitudes and practice of physicians towards fertility and pregnancy-related issues in young BRCA-mutated breast cancer patients. Lambertini M et al. Reprod Biomed Online. 2019 May;38(5):835-844).
Authors should explain the limit chosen for reproductive age (<40 years-old)><40 years) as it is not the standard and is questionable given the current availability of reproductive techniques proven to be safe in BRCA mutated women (Safety of assisted reproductive techniques in young women harboring germline pathogenic variants in BRCA1/2 with a pregnancy after prior history of breast cancer. Condorelli M et al. ESMO Open. 2021 Dec;6(6):100300).
For the quantitative survey, a total of 623 responses were received, of which 513 met the inclusion criteria (which were they?). How many surveys were sent? Response rate to be provided. Additionally, how was selected the subset of survey participants who underwent the qualitative interview?
When mentioning the informed consent authors should specify if it was written.
The socio-demo-graphic data of the quantitative survey included religion, as stated at line 122. It’s a pity such domain was not explored in more details as it might influence reproductive decision-making.
As authors claim a role for their study to enhance precision health and develop tailored interventions in the setting of germline mutation carriers, men should not be forgotten and research in this field should be envisaged.
Additionally, partners’ attitudes, potentially influencing women behaviors as well, are also not considered in both the quantitative and qualitative surveys and this should be included as a possible limitation and as another topic to be explored in future studies.
Paragraph 3.6 is quite confusing as puts together all types of risk-reducing surgery: the challenges and limitations of prophylactic mastectomy are obviously completely different from those of oophorectomy. Were data analyzed according to type of prophylactic surgery? Numbers would in principle allow to investigate if the dynamics were different according to the planned/performed surgery.
To better understand the weight of the findings from the qualitative interviews it should be reported how many women they are based on.
Another important additional descriptive piece of information, given the valuable sample size, would have been to know how many women who had children after testing positive underwent reproductive techniques and preimplantation or prenatal testing.
Author Response
We appreciate the Reviewers’ constructive comments and feel that the suggested changes have strengthened the manuscript.
The paper by Hesse-Biber and colleagues explores the complex and underreported challenges women with BRCA mutations face when considering reproduction. The sample is sizeable, and the mixed method of analysis ensures both quantitative and qualitative aspects to be valued.
- We thank the Reviewer for the positive remarks.
A few minor considerations/explanations could in my opinion improve the message delivered. A few additional papers clarifying pregnancy outcomes in BRCA mutated patients could help setting the scene and facilitating the disruption of one of the several taboos in this population of patients (e.g., Pregnancy After Breast Cancer in Patients with Germline BRCA Mutations. Lambertini M et al. Clin Oncol. 2020 Sep 10;38(26):3012-3023; Knowledge, attitudes and practice of physicians towards fertility and pregnancy-related issues in young BRCA-mutated breast cancer patients. Lambertini M et al. Reprod Biomed Online. 2019 May;38(5):835-844).
- We appreciate the reviewer bringing these two relevant publications to our attention. We have incorporated these references in the revised manuscript:
“Importantly, pregnancy is safe and there is no increased risk of adverse maternal/fetal outcomes in women with breast cancer who harbor a pathogenic BRCA variant (Lambertini et al. 2020). Further, ART does not appear to increase the risk of cancer recurrence among breast cancer survivors harboring pathogenic BRCA variants (Condorelli et al. 2021)” (Lines 83-87 or the revise manuscript).
“Although, ART does not appear to increase risk of cancer recurrence (Lambertini et al. 2020; Condorelli et al 2021), some women shared fears of increased cancer risk with IVF during interviews. In addition to patient knowledge gaps, physicians involved in cancer care also have misconceptions regarding fertility preservation and pregnancy-related issues in women with breast cancer (Lambertini et al. 2019). Thus, tailoring counselling to address such misconceptions is critical for those women who are considering pregnancy after genetic testing”(Lines 437-442 of the revise manuscript).
Authors should explain the limit chosen for reproductive age (<40 years-old)><40 years) as it is not the standard and is questionable given the current availability of reproductive techniques proven to be safe in BRCA mutated women (Safety of assisted reproductive techniques in young women harboring germline pathogenic variants in BRCA1/2 with pregnancy after prior history of breast cancer. Condorelli M et al. ESMO Open. 2021 Dec;6(6):100300).
- The reviewer raises a valid point that assisted reproductive technologies enable fertility in many women beyond 40-years of age. Our cutoff of 40 years-old was based on the reproductive endocrine literature. To address this point, we have included this in the Methods: “We defined ‘reproductive aged’ women as participants between the ages of 18-40. This cut off was based on the anticipated age-related decline in fertility as well as increased rates of aneuploidy and spontaneous abortion in women after age 40 years (ACOG 2020).“ (Lines 137-140 of revised manuscript). We also note this point in the description of study limitations: “In the present study we defined reproductive aged women as women from 18-40 years of age based on the reproductive endocrine literature. However, it is worthwhile to note that age 40 year is not a definitive cut-off for reproduction given advances in ART.” (Lines 554-557 of the revised manuscript).
We have also included the suggested reference in the Introduction: “Further, ART does not appear to increase the risk of cancer recurrence among breast cancer survivors harboring pathogenic BRCA variants (Condorelli et al. 2021)” (lines 83-86 of revised manuscript) as well as in the Discussion: “Although, ART does not appear to increase risk of cancer recurrence (Lambertini et al. 2020; Condorelli et al 2021), some women shared fears of increased cancer risk with IVF during interviews. In addition to patient knowledge gaps, physicians involved in cancer care also have misconceptions regarding fertility preservation and pregnancy-related issues in women with breast cancer (Lambertini et al. 2019). Thus, tailoring counselling to address such misconceptions is critical for those women who are considering pregnancy after genetic testing.” (Lines 437-442 of revised manuscript).
For the quantitative survey, a total of 623 responses were received, of which 513 met the inclusion criteria (which were they?). How many surveys were sent? Response rate to be provided. Additionally, how was selected the subset of survey participants who underwent the qualitative interview?
When mentioning the informed consent authors should specify if it was written.
- We appreciate the Reviewer’s comment regarding details (e.g., response rate, informed consent) for the quantitative survey. In the revised manuscript, we have the informed consent by noting “opt-in electronic informed consent” in the Methods (Line 112 of the revised manuscript). Similarly, we have noted the response rate in the Results: “For the quantitative survey, 623 online survey responses were received through patient support organizations. Incomplete surveys and those participants not meeting inclusion criteria were excluded (n=110). In total, 513 participants (82.3% of respondents) were included in the analyses.” (Lines 192-195 of the revised manuscript).
The socio-demo-graphic data of the quantitative survey included religion, as stated at line 122. It’s a pity such domain was not explored in more details as it might influence reproductive decision-making.
- The Reviewer notes an important observation regarding religion. We apologize for the oversight as we inadvertently included religion in our listing of sociodemographic in the Methods - yet we did not report any data on religion in the manuscript. As such we have removed religion from the Methods: “(i.e., age, race, marital status)” (Line 125 of the revised manuscript).
We agree that examining religiosity could be informative for understanding BRCA+ women’s reproductive decision-making. Indeed, using validated scaled to assess religiosity and fatalism/coping (i.e., Brief Multidimensional Measure of Religiousness/Spirituality, 20-item Fatalism Scale) could provide additional insights.
As authors claim a role for their study to enhance precision health and develop tailored interventions in the setting of germline mutation carriers, men should not be forgotten and research in this field should be envisaged.
- We value the reviewer’s point about sex as a variable and the inclusion of men. Indeed, we recently published a mixed-methods study examining coping response and family communication of cancer risk in BRCA+ men (Psychooncology, 2021, PMID: 34582073). In light of the Reviewer’s comment, we have noted this point in the description of study limitations: “There are several future directions that could deepen our understanding of re-productive decision-making related to BRCA carrier status. In the present study, we focused exclusively on women with BRCA variants - yet men can also carry BRCA variants. Thus, future work should include more diverse, representative samples in regards to race and ethnicity as well as sex, gender identity and sexual orientation (i.e., not just cis-gendered heterosexual women).” (Lines 558-569 of revised manuscript).
Additionally, partners’ attitudes, potentially influencing women behaviors as well, are also not considered in both the quantitative and qualitative surveys and this should be included as a possible limitation and as another topic to be explored in future studies.
- The Reviewer makes an excellent point about the role of partners and dyads in decision-making. We have noted this point in our discussion of future directions: “Similarly, attitudes of partners involved in reproductive decision making may influence reproductive decision-making. As such, future studies could examine dyadic processes as well as patient centeredness of decisional support for BRCA+ women.” (Lines 563-566 of revised manuscript).
Paragraph 3.6 is quite confusing as it puts together all types of risk-reducing surgery: the challenges and limitations of prophylactic mastectomy are obviously completely different from those of oophorectomy. Were data analyzed according to type of prophylactic surgery? Numbers would in principle allow investigating if the dynamics were different according to the planned/performed surgery.
- We thank the Reviewer for the comment regarding risk reducing surgery. As noted in the quotes in Table 4, decisions regarding risk reducing surgery were challenging for BRCA+ women. As requested, we have included numbers to provide context for the particular risk-reducing surgeries in section 3.3: “A total of 230/238 (97%) of women provided information on their specific risk-reducing surgery. Most women who underwent both mastectomy and oophorectomy had children prior to risk-reducing surgery (85/118, 72.2%, p<0.05). Only one participant reported having a child before and after oophorectomy.” (Lines 243-246 of revised manuscript). Similarly, we have added the following in section 3.6: “Of women who had a child(ren) before surgery - and who did not have a child(ren) post-surgery, most (85/134, 63.4%) had undergone both mastectomy and oophorectomy. Among women who underwent oophorectomy, 17/35, (48.6%) had a child(ren) before surgery or did not have any children (17/35, 48.6%). Only one woman reported having a child before and after oophorectomy.” (Lines 333-337 of revised manuscript).
To better understand the weight of the findings from the qualitative interviews, how many women they are based on should be reported.
- In principle, we agree with that including code counts can be informative. However, in this mixed-methods study we used qualitative inquiry to explain and deepen our understanding of the quantitative results. Our aim was to identify emergent themes rather than quantifying participants’ qualitative responses. As such, we did not perform counts within the codebook and the requested data are not available.
Another important additional descriptive piece of information, given the valuable sample size, would have been to know how many women who had children after testing positive underwent reproductive techniques and preimplantation or prenatal testing.
- We agree with the reviewer that having a deeper understanding of ART and preimplantation genetic testing could be informative. We did not collect any data on these elements in the structured survey. We have addressed this point in the discussion of the study limitations: “In addition, ART is becoming increasingly widespread and we did not specifically assess BRCA+ women’s attitudes regarding ART or preimplantation genetic testing. Additional work is needed to understand these aspects of reproductive decision-making for BRCA+ women.” (Lines 566-569 of revised manuscript).
Reviewer 2 Report
The authors provide important insights into how BRCA mutation carrier women would decide on reproductive choices. They want to provide informed decision-making for women of reproductive age while being BRCA carriers. It is an important study, which will further benefit by including other ethnicities and women belonging to different socio-economic status
Some of the comments are:
The authors can clarify or lay out how any brca+ women will consider reproductive decision-making. For example, what do they envision the support structure would look like for women who did not have children and tested positive for BRCA loss of function.
The authors can clarify BRCA+ in the abstract. Do they mean any polymorphism or known loss functions can be stated initially? Did it matter whether the mutations were known to be deleterious? Was surgery performed regardless of the status of mutations? Were the women told about the precise nature of the mutation and its correlation with cancer?
Line 198: What number of people having surgery 318 or 330, who had biological children?
What was the response from women about having children if they knew beforehand about BRCA status? Would they reconsider their decision? Also, were the support needed by BRCA+ have been discussed with them? Did the BRCA+ women discuss that with the survey and discuss what would change their minds about having children while knowingly passing the risk to their children. How does it factor into reproductive decision-making?
Author Response
We appreciate the Reviewers’ constructive comments and feel that the suggested changes have strengthened the manuscript.
The authors provide important insights into how BRCA mutation carrier women would decide on reproductive choices. They want to provide informed decision-making for women of reproductive age while being BRCA carriers. It is an important study, which will further benefit by including other ethnicities and women belonging to different socio-economic status.
- We appreciate the Reviewer’s positive comments.
The authors can clarify or lay out how any BRCA+ women will consider reproductive decision-making. For example, what do they envision the support structure would look like for women who did not have children and tested positive for BRCA loss of function.
- We thank the reviewer for asking about what envisioned supports would be needed to respond to the needs that were uncovered in this study. We have added a brief paragraph to the Discussion to specifically respond to the Reviewer’s comment: “Based on our qualitative findings, we propose that decisional support should include three key components. First, women need clear, understandable information on cancer risk and reproductive options. We envision that a psycho-educational intervention could be a key aspect of supporting informed decisions for BRCA+ women. Second, findings suggest a lifespan perspective can be useful to tailor person-centered ‘precision’ counseling and support (Figure 2). We envision that approaches to reproductive age women focus on personal risk (i.e., rebuild trust in their body) and emotional sup-port for navigating competing cancer and reproductive decisions on a compressed timeline (i.e., ART, sperm/ova donation, IVF with pre-implantation genetic testing). For older BRCA+ women, emphasis should focus on supporting concerns for family/children, alleviating parental guilt, and provide skill-building exercises to increase confidence in intrafamilial communication of risk. Women with children may also benefit from understanding actions that can be taken to address potential risk for children who may have inherited the pathogenic BRCA Third, discussion should elicit needs, values, cultural/religious beliefs, and preferences for cancer treatments (i.e., risk-reducing interventions) and reproductive goals followed by an opportunity for reflection to support high-quality decisions that are informed and aligned with the individual’s values and preferences.” (Lines 526-542 of revised manuscript).
The authors can clarify BRCA+ in the abstract. Do they mean any polymorphism or known loss functions can be stated initially? Did it matter whether the mutations were known to be deleterious? Was surgery performed regardless of the status of mutations? Were the women told about the precise nature of the mutation and its correlation with cancer?
- The reviewer makes a good point about the specific variants. This study was conducted in collaboration with patient support groups and we included women who self-identified as having a BRCA We neither reviewed participant medical records nor genetic test results. Thus, we cannot provide any data on the deleterious nature of specific variants. To clarify, this study was intended to assess women's comprehension or understanding of their genetic test results. Rather, we sought to investigate women’s reproductive decision-making after learning their BRCA status.
Line 198: What number of people having surgery 318 or 330, who had biological children?
- We thank to review catching this typographical error. We have corrected this in the revised manuscript: “Almost two-thirds of women (330/513, 64.3%) had biological children (Table 2). Of women who had risk-reducing surgery, 67.9% (216/318) had biological children.” (Lines 206-208 of the revised manuscript).
What was the response from women about having children if they knew beforehand about BRCA status? Would they reconsider their decision? Also, were the support needed by BRCA+ have been discussed with them? Did the BRCA+ women discuss that with the survey and discuss what would change their minds about having children while knowingly passing the risk to their children. How does it factor into reproductive decision-making?
- The reviewer raises important questions about the experience of BRCA+ women who had children prior to learning their BRCA We did not specifically assess regret (i.e., decisional regret scale) in the quantitative survey. In coding the interview transcripts, we did not identify any mention of regret among women who had children before learning their BRCA status. Accordingly, we did not comment on this in the manuscript.